

# Influence of the summer deep-sea circulations on passive drifts among the submarine canyons in the northwestern Mediterranean Sea.

5    Morane Clavel-Henry[1*], Jordi Solé[1], Miguel-Ángel Ahumada-Sempoal[2], Nixon Bahamon[1,3], Florence Briton[4], Guiomar Rotllant[1], Joan B. Company[1].

[1] Institut de Cienciès del Mar, Consejo Superior de Investigaciones Científicas, Barcelona, Spain

[2] Dynamics of the Ocean and Atmosphere, Universidad del Mar, Puerto Angel, Mexico

[3] Centre d'estudis avançats de Blanes, Consejo Superior de Investigaciones Científicas, Blanes, Spain

10   [4] Ecole Nationale Supérieure de Techniques Avancées, Paris Tech. Palaiseau, France

*Correspondence to*: Morane Clavel-Henry **morane@icm.csic.es**

**Abstract.** Marine biophysical models can be used to explore the displacement of individuals in and between submarine canyons. Mostly, the studies focus on the shallow hydrodynamics in or around a single canyon. In the northwestern Mediterranean Sea, the knowledge on the deep-sea circulation and its spatial variability in three contiguous submarine canyons is limited. We used a Lagrangian framework with three-dimensional velocity fields from two hydrodynamics models to study the deep bottom connectivity between submarine canyons and to compare their influences on the particle transport. The particles represented eggs and larvae spawned by the deep-sea commercial shrimp *Aristeus antennatus* along the continental slope in summer. The passive particles mainly followed a southwest drift along the slope and drifted less than 200 km within 31 days. Two of the submarine canyons were connected by more than 27 % particles if they were released at sea bottom depths above 600 m. The vertical displacement of particles was depending on the submarine canyons, the depth and the canyon wall where particles were released and it encouraged the analyses of the particle transport by canyons instead of generalizing the dynamics. In the two hydrodynamic models tested in this study, passive drift simulation differed depending on topography. Despite being run on a coarser grid, the hydrodynamic model using finer bathymetric resolution data and adjusted to the topography seemed to better model the passive drift of particles. Those results promote that the physical model parameterization has to be considered for improving the transport studies of deep-sea species.

## 1 Introduction

Lagrangian particles coupled with three-dimensional hydrodynamic models are useful to assess the impact of ocean circulation on the drift of small elements/individuals. It allows the exploration of various scenarios of dispersal and increases the knowledge in several fields of marine systems, such as predicting the direction of oil spills (Jones et al., 2016), understanding the circulation of micro-plastics (Lebreton et al., 2012) and estimating the impact of aquaculture (Cromey and Black, 2012). Using Lagrangian particles to follow living organisms give a potential spatial dispersion that may apply during the larval cycle of crustaceans, fish and other marine organisms (Ahumada Sempoal et al., 2015; North, 2008; Ospina-Álvarez et





al., 2013). Particles in the open ocean are susceptible to be advected between locations, influenced by regional currents and mesoscale features such as eddies and meanders (Ahumada Sempoal et al., 2013). The particles without implemented behavior are called passive and are efficiently used to study inactive transport of elements. They also consist of a useful approximation of larval drifts when the ecological knowledge on early life cycle is missing. The Individual-based models (IBM)
configure each Lagrangian particles with characteristics parameterized by the modeler. In marine ecological studies, IBMs provide a representation of the potential connectivity between geographically separated subpopulations, with implications for fisheries management and conservation plans (Andrello et al., 2013; Basterretxea et al., 2012; Kough et al., 2013; Siegel et al., 2003)

Simulation of marine passive drifts provides a picture of the highly dynamic circulation in the vicinity of the submarine canyons. Influence of canyons on the transport of particles and living organisms has begun to raise interest, but only a few studies are dealing with the Lagrangian transport of particles specifically in those small-scale topographic structures (Ahumada Sempoal et al., 2015; Kool et al., 2015; Shan et al., 2013). In the northwestern (NW) Mediterranean Sea, several submarine canyons, whose heads are incising the continental shelf (i.e., Cap de Creus, Palamós and Blanes Canyons) generate
mesoscale features from the main circulation pattern (Millot, 1999) called Northern Current (hence NC). When the NC crossed a canyon, it causes a downwelling on the upstream wall and an upwelling on the downstream wall in the 100-200 m layer depth (Ahumada Sempoal et al., 2015; Jordi et al., 2006) or the opposite, at lower depths (Flexas et al., 2008). Within the canyons, near-bottom currents produce a closed circulation (Palanques et al., 2005; Solé et al., 2016), which loses strength with the depth (Flexas et al., 2008; Granata et al., 1999). The canyon's shape enhances the downwelling of water
and its components (sediment and organic carbon; Puig et al. (2003)), biogenic or inorganic particles (Granata et al., 1999)), and it provides favorable and diverse habitats for benthic species (Fernandez-Arcaya et al., 2017). Those exchanges proceed at different velocity rates following the condition of the waters. For example, cascading of winter water masses drives suspended substances along the funnel structure of the canyons (Canals et al., 2009). In summer, the stratification of the water column is well-established in the NW Mediterranean Sea decoupling the mixed layer to the rest of the water column (Rojas
et al., 1995) and slowing the downward sink of biogenic particles (Granata et al., 1999). During that season, the circulation of the NC is shallower (250 m deep), wider (50 km wide) and less intense than in winter (Millot, 1999).

Up to date, the impact of the circulation on the dispersal of particles was estimated in shallow waters above the NW Mediterranean submarine canyons with a special interest on Blanes Canyon (Ahumada Sempoal et al., 2015; Granata et al., 1999;
Rojas et al., 1995). Neither the particle transports at a deeper layer nor how different layers are connected in the area of submarine canyons of the NW Mediterranean Sea have ever been estimated. Then, our attention has been drawn to researchers issuing that the strength of a predominant horizontal or vertical transport was not clear in a deeper layer (Granata et al., 1999). In accordance with those facts, we questioned if the deep-sea circulation mechanism in each submarine canyons of the NW Mediterranean Sea can be generalized.



In this study, deep passive drifts along the northwestern Mediterranean slope are simulated to determine the influence of the summer circulation modulated by the presence of the submarine canyons. We partly focused on the drift sensitivity to deep circulations using two sources of hydrodynamic to approach the transport uncertainties. With this study, we are expecting to provide a first insight into the deep connectivity between submarine canyons to improve the management of their grounds.

## 2 Material and Methods

The influence of the deep circulation on passive drifts among the submarine canyons of the NW Mediterranean Sea was analyzed through the dispersal of Lagrangian particles.

### 2.1 Hydrodynamic models

The sensibility of the dispersal to the circulation was generated with the use of two available hydrodynamic models, which covered the area of the submarine canyons in the northwestern Mediterranean Sea (see Fig 1). They were climatological simulations using the Regional Ocean Model System (ROMS; Shchepetkin and McWilliams, 2005), a free-surface, terrain-following, primitive equations ocean model. ROMS includes accurate and efficient physical and numerical algorithms (Shchepetkin, 2003). The two models were referred by their implementation versions ROMS-Rutgers and ROMS-Agrif (Debreu et al., 2012) for the sake of clarity. In the present paper, the ROMS-Rutgers configuration and its validations are presented while ROMS-Agrif has already been used and validated in Ahumada Sempoal et al. (2013).

The ROMS-Rutgers was forced by a climatological atmospheric forcing. The air temperature, shortwave radiation, long-wave radiation, precipitation, cloud cover, and freshwater flux used to force the model came from the ERA-40 reanalysis (Uppala et al., 2005). Surface pressure came from the ERA-Interim reanalysis (Dee et al., 2011). All these variables had a spatial resolution of one degree and a time resolution of 6 hours. QuikSCAT blend data was used for wind forcing (both zonal and meridional). The wind had a spatial resolution of a quarter of a degree and a time resolution of 6 hours. The boundary conditions were obtained from NEMO (available from http://www.nemo-ocean.eu; these simulations were reported in Adani et al. (2011) and interpolated to the ROMS grid to define a sponge layer of 10 horizontal grid points with a nudging relaxation time of 30 days. This methodology for the implementation of the model in the area followed the same procedure as the one already tested in the Alboran Sea (Solé et al., 2016). This model implementation was already used in previous publications as a hydrodynamic model coupled with a fisheries model (Coll et al., 2016).

The simulation domain ranges from 0.65° W to 6.08° E and from 38° N to 43.69° N. The grid resolution is 2 km (with 256 x 384 grid points horizontally) and the vertical domain is discretized using 40 vertical levels with a finer resolution near the surface (S-coordinate surface and bottom parameter $\theta s = 5$ and $\theta b = 0.4$). Thus, the thickness of the near-bottom layer on the continental slopes delimited by the 500 m and 800 m isobaths is 24 m (± 3.2 m). The advection scheme used in our simula-





tions was MPDATA (recursive flux corrected 3D advection of particles (Smolarkiewicz and Margolin, 1998)) and Large-McWilliams-Doney (LMD) mixing as a sub-grid scale turbulent mixing closure scheme (Large et al., 1994), also known as the K-profile parameterization (KPP) scheme. The air-sea interaction used for the boundary layer in ROMS was based on the bulk parameterization of (Fairall et al., 1996). We ran the ROMS-Rutgers model version 3.6 using climatological atmospher-

ic forcing and boundary conditions. The initial conditions to start the simulations were obtained using the same interpolated fields as the ones used for the boundary conditions for all variables. After an 8-year spin-up period with a baroclinic time-step of 120 seconds, we used the ninth year as the study period.

The regional ROMS-Rutgers implementation has been daily saved and validated. First, we compared the simulations with

the coarser model used for the initial and boundary conditions (NEMO). Second, the model was compared to the ARGO floats vertical data profiles of temperature and salinity to test the correct structure of the simulated water column within the year. For the ARGO tests, we selected 1900 casts in the area from October 2003 to December 2012. The data collected covers depths from the surface down to a maximum of 720 m. The ARGO vertical profile resolution was 5 m from the surface to 200 m and 25 m from 200 m to 720 m. We grouped the ARGO casts by months and by six subareas dividing the domain

according to the Coriolis force divided by the bottom depth (Fig 2). Then, we selected the subareas that have more than 30 ARGO profiles for each month, we calculated a monthly averaged profile of ARGO, and we compared them with the model climatologic profile (monthly average).

The main general behavior of the ROMS-Rutgers model simulation was coherent with the lower resolution model (NEMO),

and with the reported hydrography of the northwestern Mediterranean Sea. The model also successfully reproduced the main seasonal behavior of the different water masses. Taylor diagram (Taylor, 2001) was used to display the correlation, Root Mean Square Error (RMSE) and Standard Deviation (SD) between the monthly averages of ROMS-Rutgers model profiles and ARGO profiles by subareas (Fig 3). The comparison showed reasonably good correlations of statistical significance. For temperature, the correlation between model and ARGO was higher than 0.7 during 10 of the 12 months, while for salinity, it

was higher than 0.95 for all the months. As it is shown in Fig 3, during the month of September, the correlations in the Taylor diagram for temperature and salinity were higher than 0.9 in both variables. The month of September was selected because the thermocline in our domain disappears and the mixing process at that period of the year is likely to vary the most. Consequently, it shows that the circulation over different depths has been well reproduced in the model.

The second set of velocity and thermodynamic fields was provided by ROMS-Agrif built and validated in Ahumada Sempoal et al. (2013). The simulation domain ranges from 40.21° N to 43.93° N and 0.03° W to 6° E. It has a finer horizontal resolution (~1.2 km), with 32 sigma levels of a finer resolution near the surface (S-coordinate surface and bottom parameter $\theta s = 7$ and $\theta b = 0$) than ROMS-Rutgers. The average thickness of the near-bottom layer is 54 m and is approximately two times thicker than the near-bottom layer from ROMS-Rutgers. The model was one-way nested from a coarse resolution





model of 4 km. Bathymetry was derived from the ETOPO2-arcminute model from Smith and Sandell (1997). Surface forcing and initial and boundary conditions were built with the ROMSTOOLS package (Penven et al., 2008) before running a ten-year simulation of the model.

In ROMS-Rutgers configuration, the bottom circulation followed a southward direction along the bottom slope, except in the area south of Blanes canyon (Fig 4). Over the bottom floor of the continental slope, the highest velocities reached 8.5 cm/s on the southern mouth of Palamós canyon. The fastest vertical velocities were at sea bottom deeper than 400 m and are mostly located among the submarine canyons. The minimum and maximum velocity values (-1.4 mm/s and 1.7 mm/s) were located in Palamós canyon. In ROMS-Agrif, the hydrodynamic model had higher near-bottom currents in the areas at north

and south of Palamós canyon. Maximum velocity reached 11.6 cm/s off the continental slope between Palamós and Blanes canyons. Higher intensities of vertical near-bottom current were also in those areas (Fig 4) even though the most extreme values of vertical velocity were -1.23 mm/s and 1.11 mm/s in the bottom off the continental slope.

## 2.2 Practical study

In this study, we considered some characteristics of the *Aristeus antennatus* deep shrimp for the initial configuration of the

simulated passive drifters. First of all, we were interested in *Aristeus antennatus* because it is a benthic species distributed over the area of study (Demestre and Lleonart, 1993; Sardà et al., 1997). The population is exploited by bottom trawl fisheries and is an important income for the fishing harbors of the NW Mediterranean Sea (Gorelli et al., 2014). The good management of the fishery has raised the interest of science to have a better knowledge of the species ecology. The spawning of *A. antennatus* occurred in summer and early life stages eggs and larvae can be represented as the passive particles. Second,

the drift of particles is used as a first step to approach the unknown larval cycle. The ecology of the early life stages is hardly known due to the difficulty to catch the larvae in open-sea and to maintain alive the adults in captivity. Using the passive drift simulations, we started to explore the simplest hypothesis around the larval ecology, which corresponds to the absence of larval behavior at eggs and larval stages like the egg buoyancy or swimming abilities. Last, the particles are used to estimate potential connections between the fishing grounds of *Aristeus antennatus*. It contributes to complement the research

made in parental genetic studies (Sardà and Company, 2012), which has shown the high dispersive capacity of *A. antennatus* without revealing its connectivity paths.

For configuring the Lagrangian drift, we considered the release of particles from the depth of aggregated mature spawners of *Aristeus antennatus,* delimited by the isobaths of 500 and 800 m, i.e., in average 650 m (Sardà et al., 1997; Tudela et al.,

2003). Particles were randomly and uniformly released in the delimited zones, giving a homogeneous distribution near the bottom during the spawning peak period between July 1 and September 1 (Demestre and Fortuño, 1992) and at night (Schram et al., 2010).





For setting the duration of drift, we used the Pelagic Larval Duration (PLD) of the shrimp; or the time a larva spends in the water column from spawning to the first post-larval stage. As the superfamily Penaeidae (which contains the species of *A. antennatus*) releases eggs in the environment before hatching (Dall, 1990), the PLD definition was extended to account for the duration of the embryonic stage. To overcome the situation of an unknown PLD for *A. antennatus*, we fitted a linear model on the relation between the duration of eggs developing into post-larvae from 43 Penaeid species by reviewing research articles (see Table S1) and the temperature of the water in which the larvae were reared. The linear models, whose initial assumptions were verified (see Fig S1) and had a coefficient of determination $R^2 = 56$ %, was the following:

$$PLD = 64.71 * \exp(-0.06 * T) \ (1)$$

where $T$ is water temperature in degrees Celsius. Then, the effect of water temperature on the duration of the larval stage had the shape of an exponential law (O'Connor et al., 2007; Pepin, 1991). When particles were at the deep sea bottom, we could estimate their PLD to be 31 days because the seawater temperature was approximately 13.2 °C.

## 2.3 Biophysical model

Particle dispersal was simulated with the open-source IBM code Ichthyop (Lett et al., 2008) version 3.3, which is a free user-friendly toolset used in numerous individual dispersal studies in the western Mediterranean Sea (Ospina-Álvarez et al., 2013; Palmas et al., 2017). Displacements of virtual particles are computed by integrating the differential equation using a Runge-Kutta 4th order (RK4) advection scheme:

$$dX = U.dt \ (2)$$

where *dX* is the three-dimensional displacement vector of the particles during a time step *dt* of 30 minutes under the velocity vector *U* from the hydrodynamic models. RK4 is a stable and reliable multi-step method for numerical integration (North et al., 2009; Qiu et al., 2011), and is commonly used in dispersal Lagrangian models (North et al., 2009; Chen et al., 2003; Paris et al., 2013). The Ichthyop algorithm used tri-linear and linear interpolations in space and time from the daily average ROMS velocity fields output to each particle position at all-time steps. Attributes (geographical coordinates and depth) of each particle were saved in an output file at a daily time step.

To reach the optimal number of released particles, we carried out an analysis following a slightly modified protocol proposed in Hilario et al. (2015) and Simons et al. (2013). The drifts of 200000 advective-only particles were carried out over 31 days (PLD computed in the above section), released at midnight for each day of July and August at the bottom. A subset of N particles (N=1000, 2000, 5000, 10000, 25000, 50000, 75000, 100000 and 150000) was randomly chosen among the 200000 originally released. Then, relatively to the N particles released, the density of particles in each cell of a two-dimensional (2D) horizontal grid with 2 km resolution (i.e., density matrix) was computed. This operation was replicated





100 times, leading to a sample of one hundred 2D horizontal density matrices for a given N. The difference between the 100 replicates of that sample was evaluated by calculating the two-by-two Fraction of Unexplained Variance (FUV):

$$FUV = 1 - r^2 \quad (3)$$

where $r^2$ is the squared Pearson linear correlation coefficient within the density matrices. Based on the results, the number of particles showing a FUV lower than 5 % rounded at 20000 units. This number was set as the reference number of particles to be released per event.

## 2.4 Dispersal analysis

Seven zones were defined along the northwestern Mediterranean Sea continental slope including the Cap de Creus, Palamós, and Blanes submarine canyons (CCC, PC and BC, respectively) and the open slopes (OS) between them (see Fig 1). Those zones were also drawn to incorporate the continental shelf and the beginning of the abyssal plate as large polygons. Resulting from the prior knowledge of adult *Aristeus antennatus* and the previous analyses, we released 20000 particles in those areas on July 1 and 15, August 1 and 15, and September 1.

The contribution of hydrodynamics to particle dispersion was explored through their horizontal and vertical displacements. The horizontal displacements drifted by particles (called drift distances) were the sum of the traveled distances between the daily-recorded positions of particles. A Student two-sample test was computed to assess if the difference between those values according to a given factor (e.g., release zone, canyons, and hydrodynamics models) was significant.

In this study, the last position of the particles was attributed to the zone beneath the particle. The proportion of particles released from a release zone reaching a settlement zone was displayed in a connectivity matrix. Each cell represents the proportion of particles $P_{i,j}$ from a zone $i$ that has settled into a zone $j$:

$$P_{i,j} = N_{i,j}/N_i \quad (4)$$

where $(i, j)$ in [1:10], and where $N_{i,j}$ represents the number of particles settled in the zone $j$ which has been released in zone $i$ and $N_i$ is the number of particles released in zone $i$. Retention proportions are assumed to be the ratio of particles that remained in the zone where they were released ($P_{i,i}$ with $j=i$) and appear on the diagonal of the matrix.

## 3 Results

Dispersal of particles in both configurations of ROMS had common general patterns and mostly diverged on the transport magnitude.

## 3.1 Lagrangian dispersal within the ROMS-Rutgers outputs



Releases in canyons made generally the drift distance larger and variable (Fig 5). Lagrangian simulations carried out with the ROMS-Rutgers model transported particles over 27 km on average and up to 111 km for the longest trajectory. First, from the canyons, passive particles drift 33 km, 6 km more than the average distance, while from open slopes, particles drift 25 km, which is 3 km less than the average distance. Besides, the highest average drift distance is 36.3 km (± 15.1 km) from the

releases in the PC. Second, when released on the northern walls of the canyons, the particles drifted on 29 km, which is 10 km less than the particles released on the southern walls. Last, the open slope 1 (OS 1) and OS 3 are the release zones with the shortest transports from which particles drifted 15.2 km (± 5.5 km) and 15.7 km (± 6.8 km), respectively.

Particle retentions on the release zones were high and particle from canyons generally seeded the nearby zones (see Fig 6).

The overall retention rate was averaged at 60 % (± 33 %) because 42 % to 100 % of the particles stay in their release zones, except for the particles from the OS 2 (99 % particles drifted into the PC). Particles released on the CCC can drift up to 28 % and 27 % in the southern zones of OS 2 and PC, respectively. The connectivity of CCC with PC is possible for particles having drifted approximately 50 km (± 7 km) from 624 m (± 92 m) depth. The zone OS 3 has received 56 % of particles from the PC and has retained 96 % of particles released onside. Last, we observe a drift direction opposite to the general south-

westward direction of the main current in the south of our study area. Twenty-three percent of particles from the OS 4 drifted northward to the BC.

The vertical displacement of particles at the end of the 31-day simulation was relatively independent of the bottom depth at the release event (Fig 7). Generally, more than 26 % of particles rise above the isobath of 500 m, around 30 % particles drift-

ed without having vertically displaced, and 5 % of particles go deeper than 800 m (see Fig 7). More specifically, 12.8 % of particles released above the bottom between the isobaths of 700 and 800 m (hence, 700/800 m) arrived below 800 m. There only are 0.7 % and 4.8 % of bottom-released particles between the isobaths of 500 and 600 m and the isobaths of 600 and 700 m (hence, 500/600 m and 600/700 m), respectively, which also reach such a depth. Moreover, the downward displacement is higher for particles released in canyons than on open slopes. From the three depth layers 500/600 m, 600/700 m, and

700/800 m, as previously defined, 0.2 %, 1 % and 11.1 % of particles on open slopes and 2 %, 13.5 % and 19.7 % of particles on canyon slopes, respectively, go deeper than 800 m. Besides, the particles released on the middle layer (600/700 m) of the canyons were likely to be vertically dispersed than similar conditions of releases on the open slope (retention on the depth 27.3 % and 36.8 %, respectively).

In submarine canyons, the particles had a broader vertical displacement than in open slope. Intensity and amplitudes in the vertical displacement of particles were dependent on the canyons and their walls, especially when they were released above both walls of the Blanes Canyon (Fig 8). On average, the particle vertical displacements in canyons or above the open-slope are ascents of 28 and 24 m, respectively. Furthermore, the variability of the vertical displacement of particles in canyons (± 154 m) is 2 times higher than on open slopes (± 67.5 m). The drifts in PC have on average the highest ascent with 133 m.



The PC is also the canyon with the widest vertical displacements where the maxima upward and downward displacements are 490 m and 1163 m, respectively. From its southern wall, particles averagely rise 145 m while on the northern wall, it is an ascent of 58 m. Particles from the CCC averagely rose 44 m like for particles from the PC, but the vertical displacements were limited to a small range of 86 m in the CCC, against 170 m in the PC. The drifts from BC have in average the furthest

descent (127 m down). The drifts within BC went down by 62 m when particles are above the northern wall and go up of 36 m when particles drift above the southern wall. Contrary to the different days of release in the other canyons, the northern and southern walls of BC are where the temporal variability of the vertical displacements are the highest with a standard deviation of 53 m and 34 m, respectively.

### 3.2 Lagrangian dispersal within ROMS-Agrif model outputs

The particles were transported further in the ROMS-Agrif configuration than in the ROMS-Rutgers configuration (Fig 5) and particles from the canyon zones had traveled slightly further than particles from the open slopes. The general drift distances of particles are approximately 70.8 km ($\pm$ 26.3 km) and 44 km longer than drift in ROMS-Rutgers. Particles could drift up to the maximum distance of 197 km, detected for particles from the PC (see Fig 5). Among the canyons, particles from CCC were transported the less far away by drifting 47.1 km ($\pm$ 21 km). Oppositely, the released particles on PC averagely travel

the furthest with drifts of 84.2 km ($\pm$ 24.9 km). Particles from canyon slope zones or open slope zones were transported 73.7 km and 68.8 km, respectively, and even though the difference of drift distances is low (5 km), it is significantly different ($p <$ 0.05). Within the canyons, the particles from the northern walls of the CCC and PC drift 66 km, which was 13 km less than the particles from the southern walls. In the BC, the tendency was for particles released on the northern wall to travel 84 km, or 10 km more than particles from the southern wall.

Southern transport of particles was on a broader scope than in ROMS-Rutgers (see Fig 6). After a simulation period of 31 days, the retention rate by the release zones is 1 to 43 % of particles (Fig 6), making an average of 18 % ($\pm$ 14 %) of particles. Like in ROMS-Rutgers, the open slope OS 2 retains the least of particles (1 %). Consequently, a small proportion of 29 % of particles ends up in the PC, while the majority (64 %) connects with open slope OS 3. The particles from the CCC also

have drifted in more zones than any other release places. They were transported in four different zones, which were OS 1 at 6 %, OS 2 at 9 %, PC at 45 % and OS 3 at 19 %. Particularly, 45 % of particles connect to the PC and those particles are characterized by a drift of 70.5 km ($\pm$ 9 km) from 575 m ($\pm$ 77 m). A lower rate of dispersal is detected, as well from PC to BC. In average, 25 % of particles drift approximately 86.6 km ($\pm$ 14.9 km) from 600 m ($\pm$ 82 m) to reach BC.

The pattern of vertical particle displacements in ROMS-Agrif was similar among the different release depths and topographic structures of the NW Mediterranean Sea (see Fig 7). Overall, 37.3 % of particles reaches the layer above 500 m depth, while only 0.8 % of particles go deeper than 800 m. Particles released over the upper bottom depths (500/600 m and 600/700 m) presented comparable but still significantly different ($p <$ 0.05) vertical displacement even if they were released on can-





yons or open slopes. Around 41 % and 43 % of particles drifted within the same depth layer than their releases and 44 % to 48 % ascended in upper depths. On the contrary and overall, from releases between isobaths 700/800 m, a majority of the particles (65 %) went upward, a high proportion of particles (6.1 %) went down below 800 m, and fewer particles were transported in the depth layer of their release (retention in the deeper layer of 29 %).

The average vertical displacement of particles was a small ascent in the water column (see Fig 8), with canyons being the places of higher ascents. In average, particles rose 15 m (± 53 m) when released on open slopes, and the double; 34 m (± 65 m), when released in canyons. Among the different canyons, particles released in the PC were vertically displaced the least (23 m on average, or half the rise of particles from the CCC and BC), but the widest vertical displacements of 352 m down-

wards and 519 m upwards were from PC. We furthermore notice that the average ascent and descent of particles were the highest in BC (74 m) and in PC (44 m), respectively, which are reversed directions of movements compared to the results in the ROMS-Rutgers. Lastly, particles drifting above the northern wall of all the canyons significantly ($p < 0.05$) rise higher than for particles above the southern wall. In CCC, the rise of particles was 2.7 times higher on the northern wall (71 m) than on the southern wall, which represents the highest change of displacement between the walls. In contrast, in PC, the rise of

particles reaches 68 m on the northern wall, which was 4 m less than for particles on the other wall. Over the different release dates, the southern wall of PC is where the vertical movement of particles temporally varies the most (standard deviation of 39 m).

## 4 Discussion

Two hydrodynamic models allowed testing the sensibility of particle dispersals to the Mediterranean Sea thermohaline circu-

lation. The dispersal rates were low and similar to observations or simulations of deep drifters from other marine ecosystems (Palmas et al., 2017; Corell et al., 2012).

For the first time in the Mediterranean Sea, the connectivity between submarine canyons has been demonstrated through the Lagrangian transport of deep particles. The southward circulation which has an important amplitude on the surface also pre-

dominated at the bottom. The submarine canyons were areas that received the drifters from upstream release, as well as seeded the nearest downstream open-slopes. Previous studies already established that the flow was following the right-side canyon walls (Flexas et al., 2008; Palanques et al., 2005; Granata et al., 1999; Jorda et al., 2013) but flows and particles were not related. In this study, the drifts provided another aspect of the canyons of the NW Mediterranean Sea, as not only a channel for downward movements of terrestrial origin or sediment into deep water (Palanques et al., 2005) but also a supplier to

the neighbor zones. Each canyon has a singular topography and a different angular exposition to the main circulation, resulting with different patterns of particles dispersals. Drifts were not similar because they were not released at the same depth, the same day and at the same exposure to the current (upstream/downstream walls) inside the submarine canyons although their influence led to the common patterns explained above. Except for release on the upstream part of Blanes Canyons in





the ROMS-Rutgers, the vertical displacement of particles tended to go upward. Those vertical directions were in contradiction with the previous analyses on flow circulation above the upstream canyons (Ahumada Sempoal et al., 2015; Flexas et al., 2008; Granata et al., 1999; Puig et al., 2017). However, dispersals were simulated from deeper layers than those presented in the previous papers. The different trajectories of dispersal according to the depth and position of releases were expected

(Ross et al., 2016) and they rather reflected the unlinked currents occurring inside a canyon (Jorda et al., 2013). The internal circulation inside the canyons is more dynamic over time (Ahumada Sempoal et al., 2013; Palanques et al., 2005; Jorda et al., 2013) than the circulation along the slope. In particular, vertical drifts in PC were more impacted than vertical drifts in the other canyons.

Over the summer releases, canyons influence on dispersals had changed, even though the main patterns of connectivity persisted. In Kool et al. (2015), the main circulation was enough consistent for not impacting strongly the dispersals. This hypothesis applies for our results, even though the range of releases is over 2 months. Nonetheless, further studies need to be done on the circulation changes or the temporal drift variability inside the canyons over a regular time-series. The morphology of the canyons facilitates the retention of pelagic particles for few days (Ahumada Sempoal et al., 2015; Rojas et al.,

2014), but particle dispersals are modulated by other features in the slope morphology, particularly in dispersals in ROMS-Rutgers. The presence of a submarine valley in the slope (~3.25º E, ~41.5º N) lowered the dispersal rates and acted as a wall, which made distinct dispersals depending on the side on which particles were released. Because the submarine valley is localized between two areas of high fishing effort, an interest rises on its potential role for the marine populations. Literature categorizes their influence as canyons influence, but due to the exposure of the submarine valley to the current, an evaluation

of whether the valley deviates or recirculates the flow is needed.

In our study, the large amplitude of dispersals in ROMS-Agrif was the main origin of differences between the drifts. Particle retention was 3.3 times higher with the simulation carried out with ROMS-Rutgers than with ROMS-Agrif. Whether the particles were released on canyon slopes or on open slopes did not change the fact that distance drifts were 2 times longer in

simulations from ROMS-Agrif. The largest difference in amplitude of drifts (an average difference of 64.2 km) was simulated from the open-slope between Palamós and Blanes canyons. As mentioned above in this section, topography has perturbed the drifts in ROMS-Rutgers. In contrast, on the open-slope between the Cap de Creus and Palamós canyons, the continental slope did not have specific topographic structure, and in both models, the zone retention was low. Thus, the submarine valley influenced the circulation during the run of the two models according to the resolution of the bathymetry chosen by the mod-

elers. It was expected that adequate bathymetry would approximate better the mesoscale structures of the water circulation (Gula et al., 2015). Then, it made the simulated drifts from the ROMS-Rutgers more appropriate for the deep passive drifts even though it has a lower resolution than the ROMS-Agrif. Another major difference within the drifts in the two hydrodynamic models was the northward drifts from Blanes Canyon and from the open-slope at its south (OS4) in ROMS-Rutgers. The northward direction against the main southward circulation is a consequence of an anticyclonic flow generated at the



mouth of Blanes Canyon and also affects the deep-waters (Ahumada Sempoal et al., 2013; Jorda et al., 2013). The absence of this reverse flow in ROMS-Agrif may be the consequence of the hydrodynamic model's parameterization. The drift uncertainties gave arguments for choosing a hydrodynamic model over another, above all when drifts are in deep-water. The parameterization of the hydrodynamic model with sigma layer generally emphasizes the resolution near to the surface, relaxing the circulation variability near the bottom. In the absence of more hydrodynamic models, we encourage the deep-sea Lagrangian modeler to cross-validate the Lagrangian drift with the existing literature. This work evidences the importance of two major sources of uncertainty in the particles drift from the hydrodynamic models: bathymetry and model parameterizations. Such two factors revealed particularly the importance in the zones near the canyons (all the water column) and open-slope areas. This is an important outcome for modeling studies. In this sense, this study shows the areas where uncertainty grows and so helps to evaluate when and where the choice of the hydrodynamic models will have a minor impact in the drift results.

Eventually, associating the eggs and larvae of *Aristeus antennatus* with passive particles comprised the first approach to their drift. The duration of simulation corresponded to the first PLD approximation of *Aristeus antennatus* larvae based on Penaeid larvae. The duration appeared to be shorter than PLD for other deep-sea species (Arellano et al., 2014: Etter and Bower, 2015; Young et al., 2012) but was in agreement with PLD for temperate invertebrates (Levin and Bridges, 1995; Thatje et al., 2005; Williamson, 1982) and PLD of Penaeid species predicted in Dall (1990). Our study revealed interesting dispersal features, which could be related to the ecology of *A. antennatus*. First, few particles in canyons were transported near 1000 m depth where peaks of juvenile abundances have been reported at the end of fall (Sardà et al., 1997; D'Onghia et al., 2009; Sardà and Cartes, 1997). It would imply that the late larvae are helped or restrained by the vertical circulation to settle in the deep zones. Second, the low dispersal rates from the canyons highlighted that inactive larvae may be retained by those topographic structures. In that case, it would mean that subpopulations of *Aristeus antennatus* strongly depend on their own and local management plans as the BOE (Boletín Oficial del Estado, 2013; 2018) operating on Fishing Palamós grounds, are efficient. Nevertheless, the behavior of decapod species needs to be taken into account, as it is known to interfere in the vertical position and to influence the larval dispersal (Cowen and Sponaugle, 2009; Levin, 2006; Queiroga and Blanton, 2004). Few studies on *Aristeus antennatus* larvae sampled in open-ocean (Carbonell et al., 2010; Carretón et al., 2019; Held, 1955; Seridji, 1971; Torres et al., 2013) have revealed their presence in the surface layer. (Sardà et al., 2004) and Palmas et al. (2017) both supposed that positive buoyancy of eggs partly underlined the presence of individuals in the shallowest layer. This mechanism has to be analyzed through sensibility tests in order to compensate for the lack of accurate knowledge (Hilário et al., 2015; Ross et al., 2016).

## 5 Conclusions

We compared passive drift simulations of particles within two hydrodynamic models (ROMS-Rutgers and ROMS-Agrif). They were released according to the characteristics of the deep-sea red shrimp *A. antennatus* in a highly disturbed environ-





ment such as the submarine canyons in the northwestern Mediterranean Sea during summer, the free-living period of early stages of the species. We underlined a relatively weak dispersion of passive particles released near the bottom during the 31-day simulation. The particles drifted in southwestern adjacent zones according to the near-bottom current and according to the bottom topography. Besides the fact that particles dispersed on relatively short distances, the submarine canyons of the

NW Mediterranean Sea could be connected with deep-sea particle drifts. The amplitude and the variability of vertical displacement (upward or downward in the water column) are higher in the main submarine canyons than on other parts of the continental slope. Moreover, the vertical advection of particles in canyons was affected by their spatial distribution on the canyon walls. Our study joined numerous other studies in concluding that marine topography features, such as the deep submarine canyons and submarine valleys, favored the reception or retention of particles, as with our passive particles released

near the bottom. In deep-sea passive drift simulation, even if the hydrodynamic models have a coarse resolution (i.e., 2 km), having a finer bathymetry grid and a finer vertical resolution near the bottom, as ROMS-Rutgers, are important to have more adequate passive dispersion. Indeed, the degree of interaction between the circulation and the geomorphological structure was accounted for in the dispersal of the particles. Within the ROMS-Rutgers model, interactions between particles and the canyons were more evident than drifts made with ROMS-Agrif. Therefore, the ROMS-Rutgers model was preferred for bet-

ter modeling the bottom drifts and will be used in our future works on the benthic and deep-sea species *Aristeus antennatus*. Numerical simulations of the passive particle drift allowed the suggestion of an approximate larval life cycle of the deep-sea red shrimp *A. antennatus* and contribute to explain the effect of deep-sea circulation on some larval stages. For instance, the larvae could be drifted at deeper-depth where the recruitment occurs within the submarine canyons. However, to better represent the connectivity between canyons, further validation of the pelagic larval duration and settlement in the vicinity of the

final particle position is required, along with simulations including biological behaviors (vertical migration, buoyancy, and swimming abilities) of particles.

**Code and data availability**. The outputs (ROMS, script for Lagrangian model parameterization and scripts for analysis) presented in this article are available from the first author on request.

**Author contribution.** Morane Clavel-Henry carried out most of the research from the Lagrangian Drift simulations to the analysis of the outputs. Jordi Solé developed and validated the ROMS-Rutgers model. Miguel Angel Ahumada used his ROMS-Agrif model output to simulate Lagrangian drifts. Florence Briton provided codes for analyzing the Largangian drift sensibility. Guiomar Rotllant and Joan Baptista Company guided the first author with the biology of the red shrimp. Joan

Baptista Company is the principal researcher of CONECTA's project and has provided the scholarship to M C-H to realize this study. M. C-H prepared the manuscript with contributions from all co-authors.

**Competing interests**. The authors declare that they have no conflicts of interest.





**Acknowledgements.** The authors greatly thank for the time and support of M. Carretón and the computer technicians from the CEAB at Blanes and ICM at Barcelona. The authors also appreciated the numerous comments, which helped to structure the article.

**Fundings.** Fundings were provided through the CONECTA's project supported by the Ministerio de Economia, Industria y Competividad from Spain Government. M. C-H is funded under FPI PhD program of the Spanish government.

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



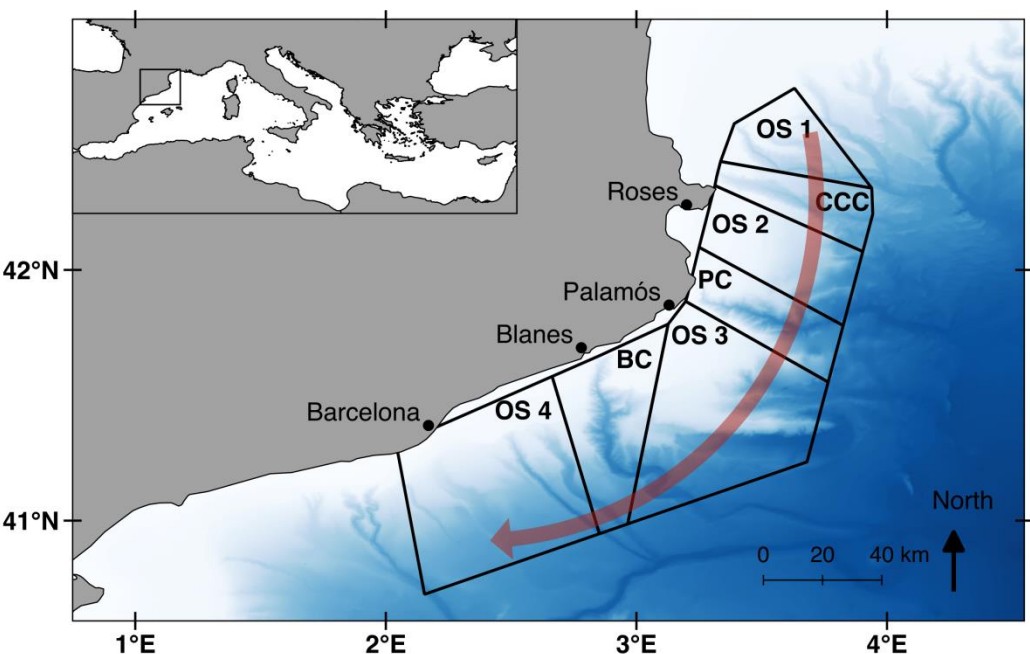

**Figure 1. Maps showing the Mediterranean Sea with the study area (top left map) and the general topography in the study area. Release zones are represented along the northwestern Mediterranean Sea: Open Slope (OS) 1, Cap Creus Canyon (CCC), OS 2, Palamós Canyon (PC), OS 3, Blanes Canyon (BC) and OS 4. Red arrow indicates the general direction of the Northern Current. Topographic layer was extracted from EMODnet (http://www.emodnet-bathymetry.eu) and the Mediterranean Sea coastline was provided by the public domain map dataset made with Natural Earth (http://github.com/nvkelso/natural-earth-vector).**



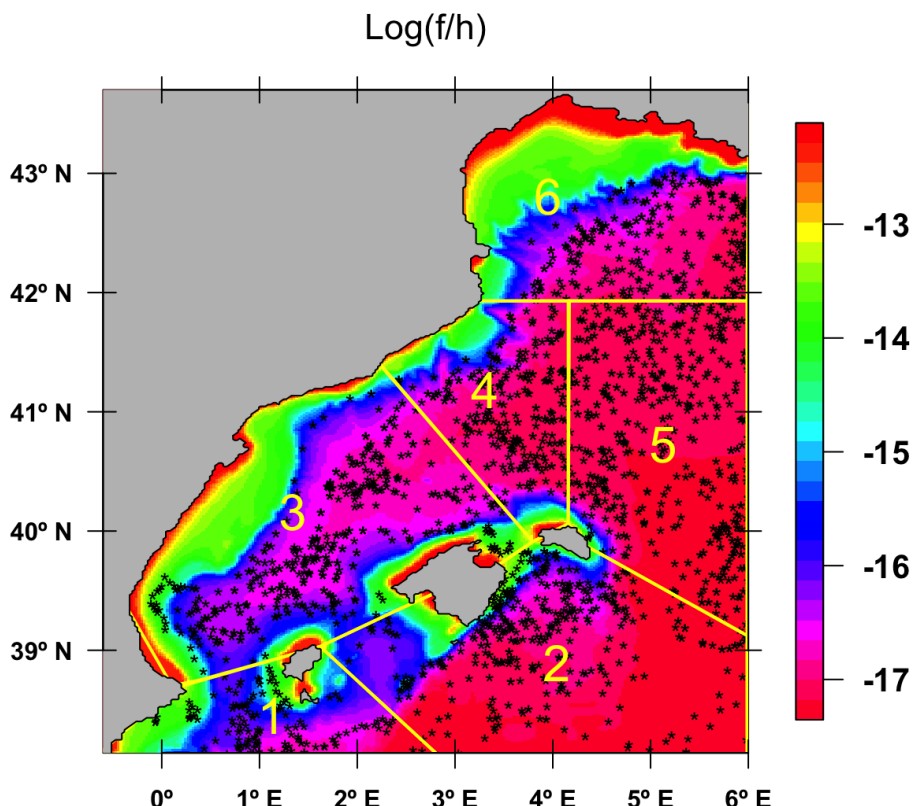

**Figure 2. Map of ROMS-Rutgers model domain and position of ARGO buoys. The color panel represents the logarithmic-scaled Coriolis parameter (f) divided by the depth (h) (f/h; zebra color palette), which were gridded in the ROMS-Rutgers model. Then yellow polygons indicate six subareas created by the modelers to validate the model using ARGO float observations. The black dots in the domain represent all the ARGO profiles used for validation analyses.**





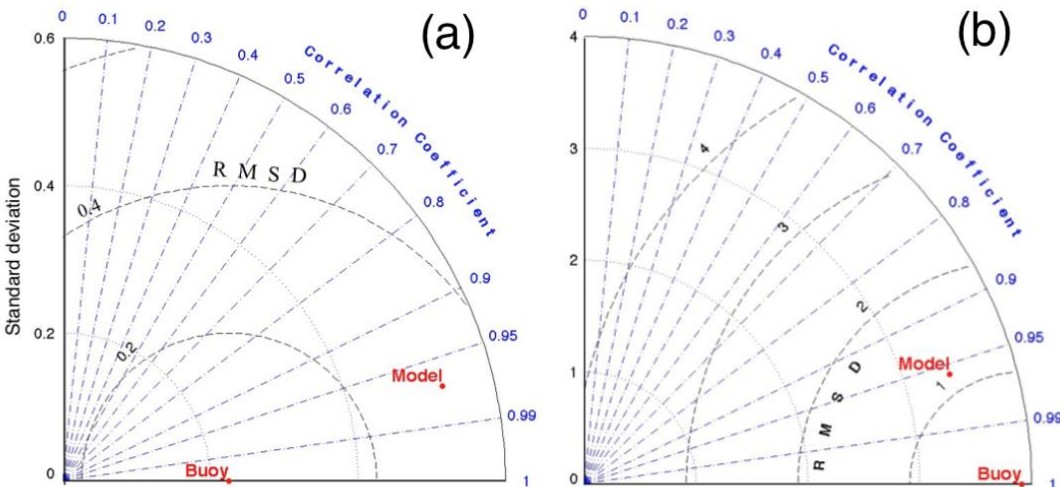

**Figure 3. Taylor diagrams for validation of ROMS-Rutgers. Example of comparison between buoys (ARGO floats) and ROMS-Rutgers model simulations during September. (a) The temperature of float versus model and (b) the salinity of float versus model. Radial from the origin (0, 0) is Correlation Coefficient, intern circle from origin is standard deviation, intern dashed circle centered by the buoy position is the Root Mean Squared Distance (RMSD).**





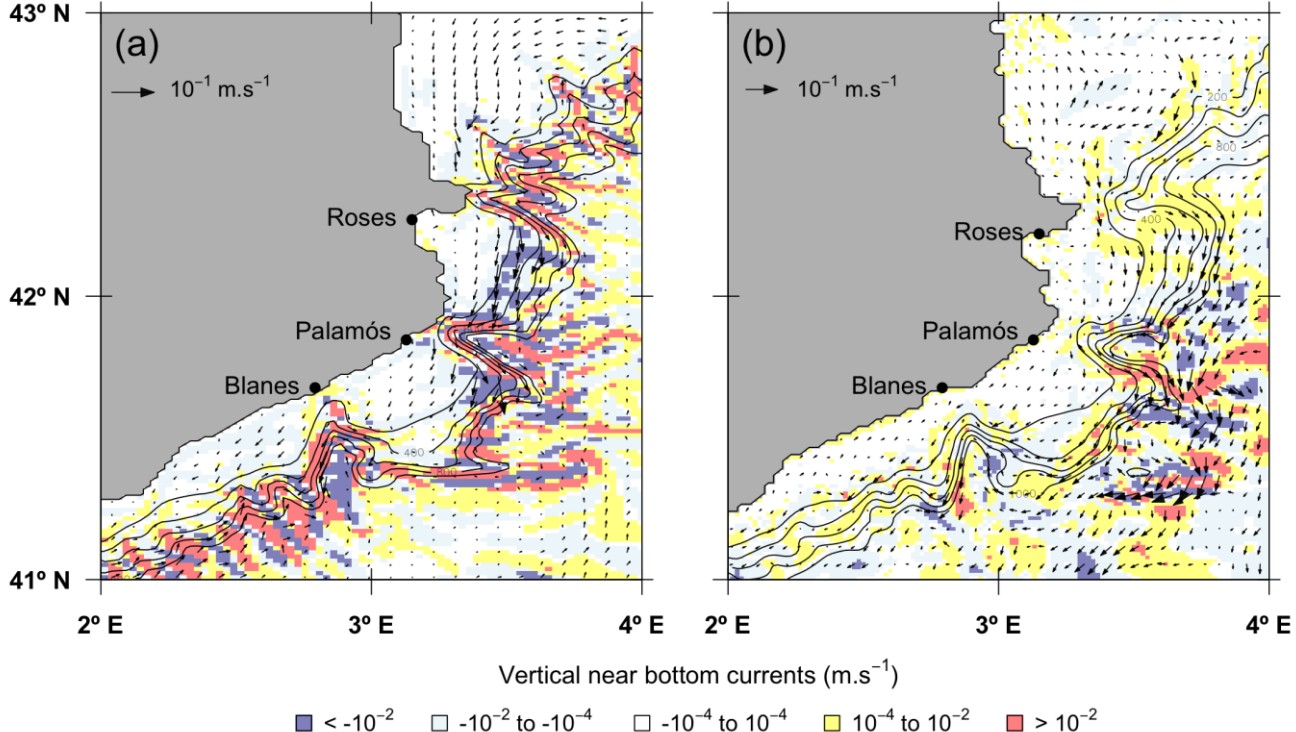

Vertical near bottom currents (m.s⁻¹)

$< -10^{-2}$   $-10^{-2}$ to $-10^{-4}$   $-10^{-4}$ to $10^{-4}$   $10^{-4}$ to $10^{-2}$   $> 10^{-2}$

**Figure 4. Near-bottom vertical (colors) and horizontal (arrows) velocities in the northwestern Mediterranean Sea extracted from the (a) ROMS-Rutgers and (b) ROMS-Agrif hydrodynamic models. Blue colors are for downward (negative) vertical velocities. Yellow and red colors are for upward (positive) vertical velocities. White colors are for small vertical velocities (from $-10^{-4}$ to $10^{-4}$ m.s⁻¹). Continuous black lines are the isobaths at 200 m, 400 m, 600 m, 800 m and 1000 m adapted from the gridded bathymetry in each hydrodynamic models.**





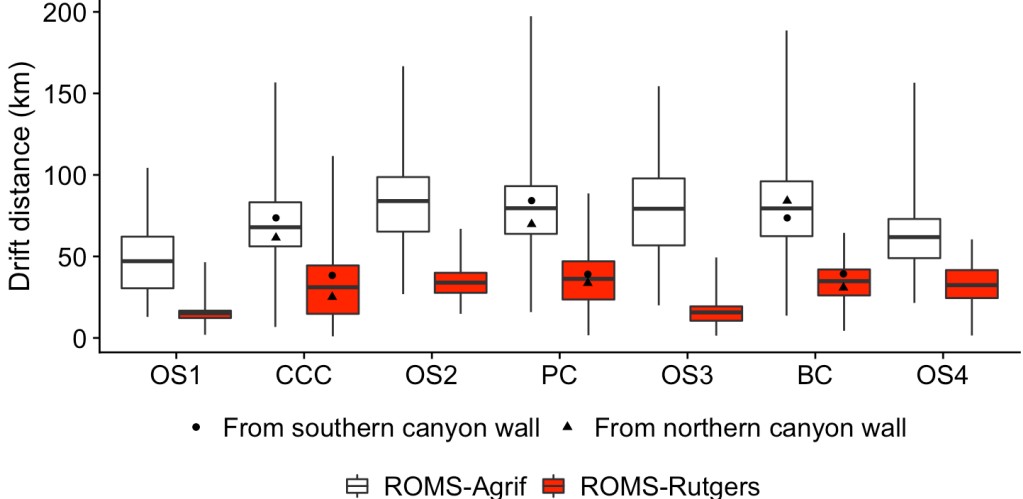

**Figure 5.** Horizontal transport of passive particles (km) from the release zones in the ROMS-Rutgers (red bars) and ROMS-Agrif (white bars) simulations. In the canyons (CCC, PC, BC), the average transport has been calculated for the particles from the northern (full triangle) and southern (full circle) walls. Boxplots represent the minimum, quantile 25 %, mean, quantile 75 % and maximum. Release zones code indicated in Fig 1.





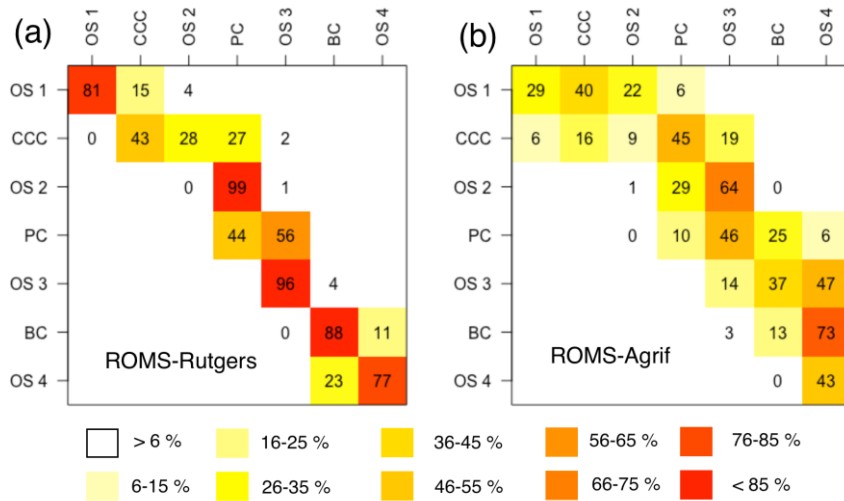

**Figure 6. Dispersal rate (in percentage %) of passive particles between the zones of release (Y-axis) and settlement (X-axis) from particle drift simulations in Roms-Rutgers (a) and ROMS-Agrif (b). The retention rates are on the diagonal. Release zones code indicated in Fig 1.**


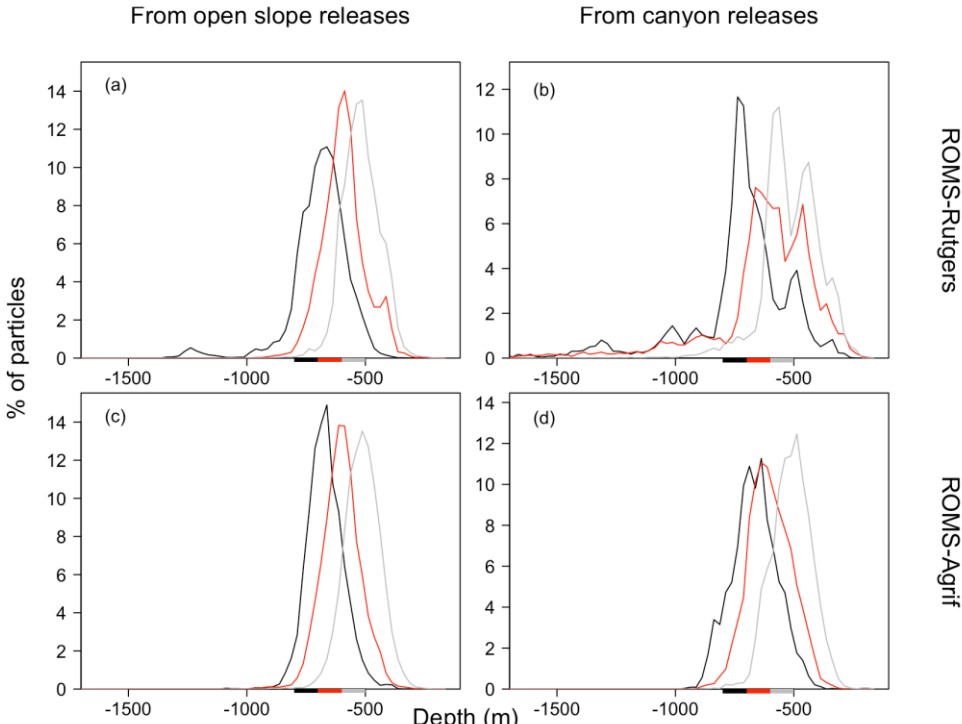

**Figure 7. Depth range (m) of particles after 31 days of drift. Particles were released between the following three depth ranges: 500-600 m (grey), 600-700 m (red) and 700-800 m (black) bars on the X-axis on open slopes in ROMS-Rutgers (a), on open slopes in ROMS-Agrif (b), within canyons in ROMS-Rutgers (c), within canyons in ROMS-Agrif (d).**



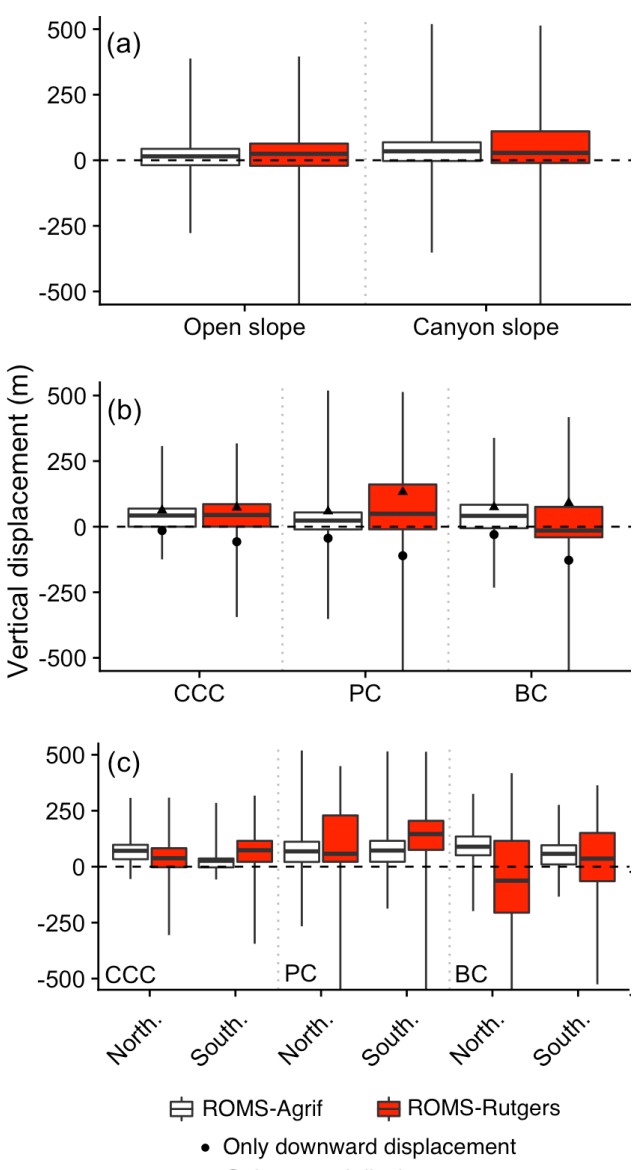

**Figure 8.** Vertical displacement of particles having drifted on the open slope or the canyon slope (a), drifted within the canyons (b), and been released on the northern wall (North.) or southern wall (South.) of the canyons (c) in the ROMS-Rutgers (red boxplot) and the ROMS-Agrif (white boxplot) simulations. In (b), the average downward and upward displacements are indicated by the full circle and triangle, respectively. CCC, PC and BC stand for Cap de Creus Canyon, Palamós canyon and Blanes Canyon, respectively. Release zones code indicated in Fig. 1.