# Peer review of "Influence of the summer deep-sea circulations on passive drifts among the submarine canyons in the northwestern Mediterranean Sea."

_Ocean Science, 2019_

## Referee Comment (RC1) · Xavier Durrieu de Madron (Referee) · 8 Aug 2019

General comments This paper is a first approach to simulate the drift of larvae of the shrimp Aristeus Antennatus that live in canyons off the Catalan coast, and thus to identify potential retention within each canyon and open slope, as well as connectivity between adjacent canyons. This is a first approach in the sense that larvae, in the absence of specific information on their dynamics, are treated as passive particles.

In addition, a comparison of the effect of the spatial resolution of the model, affecting the fineness of bathymetry and vertical resolution near the bottom.

[Figure]

The approach and results appear robust and provide interesting information on the dispersion of Lagrangian particles in this complex environment. The manuscript is clearly presented and illustrated. It deserves to be published in Ocean Science with some corrections.

In general, the discussion would benefit from the use of sub-sections defined by headings to clarify its course and make it easier to read, and a more detailed analysis of the particle trajectory and a more in-depth analysis on the trajectory and dispersion particles would be useful.

Specific comments Page 2, Lines 30-31. The articles by Durrieu de Madron, 1994 and Durrieu de Madron et al. 1999 addressed the dynamics of water masses and dispersion of suspended particles within the Grand-Rhône canyon and the adjacent open slope, located upstream of the area studied in this article. Perharps they would provide some informations on the dynamics of suspended particulate matter applicable to this study. - Durrieu de Madron et al. (1999) Role of the climatological and current variability on shelf-slope exchanges of particulate matter. Evidence from the Rhône continental margin (NW Mediterranean). Deep-Sea Research, 46, 1513-1538 - Durrieu de Madron (1994) Hydrography and nepheloid structures in the Grand-Rhône canyon. Continental Shelf Research, 14, 457-477.

Page 10, lines 19-21. Could you explain in more detail why you decided to use these two versions of the model, one of which using the AGRIF grid refinement system?

Discussion part. It would be interesting to present some typical drifts showing both the trajectory and the depth reached by these particles along their path. Furthermore, the effect of the longer PLD duration on the results of this simulation could have been estimated. Is such a sensitivity analysis with the model considered to be the most efficient possible?

Technical corrections Page 2, Line 24. Is it the decoupling of the surface mixed layer from the rest of the water column?

Pages 7, line 10. I suggest replacing "the beginning of the abyssal plain" with "continental rise".

Page 12, lines 15-18. This sentence has no place here, but rather in the chapter Âń 2.2 Practical study Âż on page 6 to compare the duration of the chosen PLD with that of the literature.

Figure 6. Change the sign > 6% to < 6%

Figure 7. The lines are a little too thin (especially for blue) and could be thickened.

---

## Author Comment (AC1) · 28 Aug 2019

The author, on behalf of the co-authors, has read and thanks for the reviews and the suggestions from Xavier Durrieu de Madron (RC1). We examined all of them for inclusion in the reviewed version of the manuscript, even though some comments are still in discussion with the co-authors and no detailed among the responses.

Below, there are responses to distinct comments.

1-Headlines in Discussion will be added, in lines with the paragraph contents and the objectives of the paper.

[Figure]

2-Specific response: The author appreciates the given references regarding the exchange of particle matters, the hydrology and the nepheloid layer in a French canyon. It provides indeed information that can be compared with similar studies lead in Cap de Creus and Blanes canyons and provides additional supports of concerns with the objectives of the study.

For example: Lopez-Fernandez P, Calafat A, Sanchez-Vidal A, Canals M, Mar Flexas M, Cateura J, et al. Multiple drivers of particle fluxes in the Blanes submarine canyon and southern open slope: Results of a year round experiment. Progress in Oceanography. 2013;118:95-107

3-Discussion response: Based on the suggestions and the objectives of the manuscript, it would be interesting to represent the trajectories of particle released in the three canyons with a color guide according to the particle depth. This plot can be an illustration of the whole findings described in this manuscript.

We worked with a drift stimulation time of 31 days for complying with an estimated Pelagic Larval Duration (PLD) of the shrimp larvae, which consisted of our case studied species. Because unknown, this PLD was estimated based on a linear regression model described in the manuscript and the near-bottom water temperature, which is stable (around 13.2 °C) in our studied area. The study focused on the spatial variability in the trajectories induced by the circulation in the canyons. We recognized that exploring if longer PLD would induce significant changes in the trajectories is interesting and it introduces an analyze on the temporal dimension. We dealt with this idea in the context of another study (in preparation) implying the same species, the considered model (ROMS-Rutgers) and different objectives.

4-Figures were modified according to the suggestions of the RC1. They are enclosed with our response. Fig

[Figure]

**Fig. 1.** Figure 6

**From open slope releases**

**From canyon releases**

ROMS-Rutgers

ROMS-Agrif

% of particles

Depth (m)

**Fig. 2.** Figure 7

---

## Referee Comment (RC2) · Anonymous Referee #2 · 23 Sep 2019

This paper examines connectivity of bottom spawned larvae between submarine canyons. The value to fisheries lies in optimal scaling of management regions. The introduction sets a clear rationale for the study. The methodology section is thorough and detailed, presenting two models which will be used with varying parameterizations. The results give a clear number for connectivity.

However, I am reluctant to trust models without in-situ validation. In my experience with modeling, I am convinced that models are principally to be used as tools to guide further study. But I am also aware that the complexity of details that can be achieved by models may not presently be amenable to validation.

[Figure]

The conclusion section is rather lame compared to the rest of the paper and contains poor English phrasing. My recommendation is that the conclusions be given more effort. Since fishery managers tend to believe models, I recommend that you make some effort at dissuading them that model details are not substitutes for accuracy. For example, a 4th order Runge Kutta with added sub-grid scale turbulence is still uncertain.

---

## Author Comment (AC2) · 2 Oct 2019

Estimated Referee

Thank you for your feedback and comments on the manuscript.

I understand your reluctance to trust models, but, to date, it is still the best method available for approaching the connectivity path of deep-sea species. This manuscript is the first step of a work that is conducted with the final focus to improve fishery management. Unfortunately, modeled deep-sea dispersions are hard to validate due to lack of empirical data and it is an issue that some projects-to-be on deep-sea are plan-

ning to resolve. Nonetheless, we believe that intermediate tools such as genetics can partially validate biophysical models. Validation of them with parental genetics and genetic connectivity can be used for coastal or epipelagic species, but for deep-sea species, this effort has not been noticed yet. Those genetic data for the species-case Aristeus antennatus in the studied zone are still under investigation and could not even be provided as a preliminary result.

Following your comments about the conclusion, we will revise it for including more convincing sentences and revising the English. We will insist that this paper joins the few articles dealing with deep-sea connectivity among submarine canyons and that it provides information on which future studies will rely and step on. In that context, we will remind the readers that this paper is a preliminary work engaged in fishery management, but, as first results, the fishery managers should wait for future studies.

Upon request and further discussion with the editor, we can provide a version of this modified conclusion through the interactive comments.

Regards,

On the behalf of the co-authors,

Morane Clavel-Henry
* * *

---

## Author Response (AR1)

**Author's Response**

| Line | Reviewers | Comments | Author's response |
|---|---|---|---|
| l. 63-64 | R1 | The articles by Durrieu de Madron, 1994 and Durrieu de Madron et al. 1999 addressed the dynamics of water masses and dispersion of suspended particles within the Grand-Rhône canyon and the adjacent open slope, located upstream of the area studied in this article. Perhaps they would provide some informations on the dynamics of suspended particulate matter applicable to this study | We modified the text accordingly to the comment suggesting the studies of sediment traps and water fluxes in canyons. References have been added. l.621 and l.572

 For better including this change, the text from l.62 to l.71 has been modified. |
| l. 58 | R1 | Is it "the decoupling of the surface mixed layer **from** the rest of the water column" | Modified in the text. |
| l. 82-86 | R1 | Could you explain in more detail why you decided to use these two versions of the model, one of which using the AGRIF grid refinement system? | Material and Methods were reinforced it with additional details. |
| l. 435-438 to l.196-198 | R1 | This sentence has no place here, but rather in the chapter ´ 2.2 Practical study ˙ on page 6 to compare the duration of the chosen PLD with that of the literature. | The text has been displaced from the Discussion section to the subsection 2.2. Practical study (Methods). The subsection 2.2 was reordered for better clarity. |

| l. 237 | R1 | I suggest replacing "the beginning of the abyssal plain" with "continental rise". | Modified in the text. |
|---|---|---|---|
| Lines 367, 395, 432 | R1 | The discussion would benefit from the use of sub-sections defined by headings to clarify its course and make it easier to read | We divided and named three subsections in the Discussion section.

Changes in relation with the reviewer comments were marked-up.

Due to some texts that did not fit with the sub-section subject, pieces of text were displaced in other subsections. Most of those modifications were unmarked for showing the relevant changes suggested by the reviewers. |
| Figure 9 | R1 | It would be interesting to present some typical drifts showing both the trajectory and the depth reached by these particles along their path. | We plotted some drifts showing both the trajectory and the depth layer from which particles were released. It provides a visual of the drifts and generalizes the findings described in the manuscript. The figure was named Fig.9 (l.761) and two paragraphs were added to the manuscript at lines l 300-308 and 353 - 361 |
|  | R1 | Furthermore, the effect of the longer PLD duration on the results of this simulation could have been estimated. Is such a sensitivity analysis with the model considered to be the most efficient possible? | We worked with a drift stimulation time of 31 days for complying with an estimated Pelagic Larval Duration (PLD) of the shrimp larvae, which consisted of our case studied species. Because unknown, this PLD was estimated based on a linear regression model described in the manuscript and the near-bottom water temperature, which was established around 13.2 ºC in our studied area. The study focused on the spatial variability in the trajectories induced by the circulation in the canyons. We recognized that exploring if longer PLD would induce significant changes in the trajectories is interesting and it introduces an analyze on the temporal dimension. We dealt with this idea in the context of another study (in preparation) implying the same species, the considered model (ROMS-Rutgers) and different objectives. |

| | | | |
|---|---|---|---|
| Discussion | R1 | A more detailed analysis of the particle trajectory and a more in-depth analysis on the trajectory and dispersion particles would be useful. | We still modified the text to comply with the suggestions. Those modifications were included in the restructuration of the Discussion and the three subsections. |
| Figure 6 | R1 | Change the sign > 6 % to < 6 % | Modified |
| Figure 7 | R1 | The lines are a little too thin (especially for blue) and could be thickened. | Modified (colors and width of the lines) |
| Conclusion L459 to 502 | R2 | The conclusion section is rather lame compared to the rest of the paper and contains poor English phrasing. My recommendation is that the conclusions be given more effort. Since fishery managers tend to believe models, I recommend that you make some effort at dissuading them that model details are not substitutes for accuracy. | Conclusion was written again to include the suggestions of RC2. We improved both the weight of our results and the English phrasing.

We warned the reader about the state of our research (i.e., in this manuscript, the first time that Lagrangian dispersal was used in deep-sea). |
| Conclusion | R2 | …and contains poor English phrasing. | This comment supported a whole revision of the manuscript regarding the English phrasing. The main changes are:
l. 168-177
l. 400-402 |
| Figures 4, 5, 8 | Author | | Figure 4 has been updated with a better resolution of the images.

We improved the quality of figures 5 and 8. |
| References l.544 | Authors | | Manuscript of Carretón et al. (2019) has been published and therefore, the reference has been updated |

[revised manuscript text omitted]

Vertical near bottom currents (m.s$^{-1}$)

$\blacksquare$ < -10$^{-2}$   $\square$ -10$^{-2}$ to -10$^{-4}$   $\square$ -10$^{-4}$ to 10$^{-4}$   $\blacksquare$ 10$^{-4}$ to 10$^{-2}$   $\blacksquare$ > 10$^{-2}$

[Figure]

Figure 4. Vertical velocity and horizontal velocity vectors at bottom in the northwestern Mediterranean Sea in the (a) ROMS-Rutgers and (b) ROMS-Agrif. Colours from blue to red according to the downward or upward (negative/positive) vertical velocity. Continuous black lines are the isobaths adapted from the bathymetry of the hydrodynamic models.

740

[Figure]

**Figure 5. Horizontal transport of passive particles (km) from the release zones in the ROMS-Rutgers (red bars) and ROMS-Agrif (white bars) simulations. Boxplots represents the minimum, quantile 25 %, mean, quantile 75 % and maximum. Release zones code as indicated in Fig 1.**

[Figure]

**Figure 6. Dispersal rate (in percentage %) of passive particles between the zones of release (Y-axis) and settlement (X-axis). The retention rates are on the diagonal. Release zones code as indicated in Fig 1.**

[Figure]

[Figure]

**Figure 7. Depth range (m) of particles after 31 days of drift. Particles were released between the following three depth ranges: 500–600 m (grey), 600–700 m (red) and 700–800 m (black) bars on the X-axis.**

[Figure]

**Figure 8. Particles vertical displacement. Vertical displacement according to particles drifting on the Open Slope or the Canyon Slope (a), drifting within the canyons (b) and released on the northern wall (North.) or southern wall (South.) of the canyons (c) in the ROMS-Rutgers and the ROMS-Agrif simulations. Release zones code as indicated in Fig. 1.**

755

760

[Figure]

765

**Figure 9. Estimated transport paths of passive particles during 31 days. Passive particles were released in the Cap de Creus canyon (a and d), Palamós canyon (b and e) and in Blanes canyon (c and f) and advected by the modeled hydrodynamics from ROMS-Rutgers (a-c) and ROMS-Agrif (d-f). Colors of the transport paths was associated with the depth ranges 500-600 m (grey), 600-700 m (red), and 700-800 m (black) on which particles were released.**